# Regression Tree Analysis for Stream Biological Indicators Considering Spatial Autocorrelation

**DOI:** 10.3390/ijerph18105150

**Published:** 2021-05-13

**Authors:** Mi-Young Kim, Sang-Woo Lee

**Affiliations:** 1Graduate Program, Department of Environmental Science, Konkuk University, Gwangjin-Gu, Seoul 05029, Korea; kmy91321@konkuk.ac.kr; 2Department of Forestry and Landscape Architecture, Konkuk University, Gwangjin-Gu, Seoul 05029, Korea

**Keywords:** regression tree analysis, biological indicators, spatial autocorrelation, land use, principal component analysis

## Abstract

Multiple studies have been conducted to identify the complex and diverse relationships between stream ecosystems and land cover. However, these studies did not consider spatial dependency inherent from the systemic structure of streams. Therefore, the present study aimed to analyze the relationship between green/urban areas and topographical variables with biological indicators using regression tree analysis, which considered spatial autocorrelation at two different scales. The results of the principal components analysis suggested that the topographical variables exhibited the highest weights among all components, including biological indicators. Moran′s I values verified spatial autocorrelation of biological indicators; additionally, trophic diatom index, benthic macroinvertebrate index, and fish assessment index values were greater than 0.7. The results of spatial autocorrelation analysis suggested that a significant spatial dependency existed between environmental and biological indicators. Regression tree analysis was conducted for each indicator to compensate for the occurrence of autocorrelation; subsequently, the slope in riparian areas was the first criterion of differentiation for biological condition datasets in all regression trees. These findings suggest that considering spatial autocorrelation for statistical analyses of stream ecosystems, riparian proximity, and topographical characteristics for land use planning around the streams is essential to maintain the healthy biological conditions of streams.

## 1. Introduction

Streams and rivers are complex lotic systems witnessing continuous changes. Interactions among multiple factors, such as topography, physiochemistry, and hydraulics in watersheds directly and indirectly affect the stream condition [1,2]. Previous studies have examined the relationship between instream habitat quality and aquatic organisms [3,4,5]. As instream habitat quality is determined by watershed and riparian ecosystem conditions, landcover, topography, physical habitats, and organisms of these systems are critical factors that determine the stream water quality and biological condition [6,7]. In particular, the surrounding land use strongly influences the biological integrity of streams, and thus, multiple studies have been conducted to identify the relationship between watershed land use and biological condition [5,8,9,10,11,12,13,14,15]. Recently, watershed topography has significantly changed due to the increasing population and high concentration of industries in urban areas. Buildings, roads, and parking spaces in urbanized areas along with anthropogenic uses of watersheds for urbanization, cultivation, and other activities have altered surface water characteristics, hydraulic and hydrologic systems, and the amount of pollutant emissions and pollutant loads in watersheds. Consequently, these alterations have degraded physiochemical and ecological conditions of streams [16,17,18]. Additionally, previous studies reported that riparian forests adjacent to streams directly influence the flow mechanisms of sediments, organic matter, and nutrients from watersheds [19,20]. Riparian forests have various functions, such as preventing bank erosion, reducing organic matter and nutrients, controlling water temperature, providing habitats for aquatic and terrestrial organisms, increasing ecological health and diversity, and mitigating the negative effects of land use on rivers [19,20,21,22,23,24,25,26,27,28,29,30,31,32,33]. Therefore, the relationships between land cover (particularly, green and urban areas) and stream conditions need to be studied in detail to develop programs for improving the stream health and implementing stream restoration and management strategies [34].

As reported previously, the extent of forested areas and the width of riparian areas strongly influence biological communities [35,36]. However, specifying buffer widths is challenging because the buffer scale typically used varies depending on topography, geology, hydrology, rainfall intensity, and vegetation type [37]. Additionally, the responses of stream ecosystems to anthropogenic modification (e.g., land use, subsurface modification, groundwater abstraction, stream channelization, and damming) depends on the modification types and scales [38,39]. Thus, buffer widths might need to be determined based on the specific response (e.g., water availability, productivity, water quality, the composition of species, microclimate regulation, habitat loss, flow regime) and scale (e.g., catchment scale, landscape scale, segment scale) of the stream ecosystem. Damanik-Ambarita et al. [38] conducted an extensive review about the various riparian and catchment scales used in previous studies as a part of a study in exploring the quantitative relationship between land use and the aquatic macroinvertebrate community in previous studies. According to their study, the scale of the riparian buffer used in previous studies is ranging from 30 to 5000 m width (or radius) [40,41,42]. Other researchers have proposed multiple spatial scales and extents to identify the relationship between land use and stream ecosystem [43,44]. For example, Burdon et al. [31] used 5 m buffer scale in testing the relationships between riparian integrity with ecological status in European streams. Furthermore, in another study, the buffer width differed according to the study purpose, for example, 30 m for water quality conservation, 150 m for protection from flood, and more than 500 m for riparian habitats and biological communities [45]. Yirigui et al. [37] reported that a buffer width greater than 500 m is necessary for nutrient control, bank stabilization, aquatic and terrestrial wildlife habitat, and microclimate change mitigation. The Korean Ministry of Environment (MOE) has adopted two buffer widths, 500 m and 1 km, to preserve riparian areas and conserve drinking water quality [46]. Kim et al. [47] studied the relationship between fish and land use using multiple spatial scales in a 5 km buffer zone and reported that the effects of forest areas were higher at a high spatial scale.

Studies conducted to identify the relationship between landscape variables and biological integrity of streams have used conventional correlation and regression analyses, which statistically assume that all variables are independent [48]. However, the impacts of landscape variables on the biological communities in streams are not completely independent and instead, exhibit spatial autocorrelation [49,50,51]. Spatial autocorrelation is a phenomenon in which a variable at one point correlates with the same variable around it in a two-dimensional space [48,52,53]. Lee et al. [54] indicated that spatial autocorrelation of landscape characteristics can occur because of the continuity and hierarchical structure of environmental conditions. Therefore, spatial autocorrelation is critical because the measured environmental variables are not random but have a common structural variance [55]. As conventional statistical methods assume randomness of the measured data, using a new approach to address spatial autocorrelation is necessary [56]. However, it is noteworthy that some studies argued that local conditions and sampling methods, such as watershed-based sampling [57], and multi-distance sampling can overrule the effects of spatial autocorrelation in some cases [58].

This study aims to analyze the relationships between green spaces, urban areas, and biological integrity of streams at two different scales in Nam-Han River Basin. The objectives were (i) to identify spatial autocorrelation of biological indicators represented by diatoms, macroinvertebrates, and fish, (ii) to analyze the relationship for each biological indicator with land cover (green and urban areas) and topographical variables using two different riparian scales (5 km and 500 m), and (iii) to suggest scientific strategies for the restoration and management of green spaces and streams in the basin. The findings would thus, provide insights for improving water quality and the biological condition of streams in aquatic ecosystems.

## 2. Materials and Methods

### 2.1. Study Area and Sampling Site

The Nam-Han River Basin is located centrally in the Korean Peninsula and is a part of the major streams of the Han River, the largest water system in Korea (Figure 1). The Nam-Han River is a major river that forms the Han River System along with the Buk-Han River. It covers approximately one-third (12,577 km^2^) of the total area of the Han River System (34,473 km^2^), and its total length is 375 km [59]. The average annual rainfall in Korea is 1159 mm and that in the Nam-Han Basin is 1409 mm. Approximately 65% of the total annual precipitation occurs from June to September and the rivers are parched from March to May and October to November. The Nam-Han River Basin is located at a high elevation (more than 1000 m above sea level) [60]. Urbanized areas, including cultivation areas in the Nam-Han River Basin are concentrated downstream, and cover approximately 24% of the total area. Furthermore, forests cover approximately 70% of the total area of the basin [61].

Since 2007, the Korean MOE has been conducting a National Aquatic Ecological Monitoring Program (NAEMP) to monitor the water quality and ecological condition of streams and rivers across the country. NAEMP is significant for stream ecosystem research because it investigates and evaluates various aquatic ecosystem characteristics, such as structure, vegetation, water quality, and ecological health of rivers [62]. Additionally, the MOE has been monitoring aquatic properties, such as physiochemical water quality index, including total nitrogen (TN), total phosphorous (TP), biochemical oxygen demand (BOD), chemical oxygen demand (COD), dissolved oxygen (DO), chlorophyll a (Chl-a), and pH; biological indicators, including trophic diatom index (TDI), benthic macroinvertebrate index (BMI), and fish assessment index (FAI); and topographical characteristics, including width, depth, and velocity of the five major rivers through 800 monitoring sites [63]. In this study, we sampled 107 sites mentioned in the tributary datasets of the Nam-Han Basin in 2018 (excluding the main stream because it has complex and diverse effects apart from effects of land cover) to investigate the relationship between stream biological conditions, green and urban areas, and topographic variables. 

### 2.2. Biological Indicators of Streams

In this study, scores of three biological indexes developed by NAEMP were used as indicators of stream biological conditions in the Nam-Han River Basin. A score of 0 to 100 was assigned for each biological indicator, which were later classified into Class A (Excellent), Class B (Good), Class C (Fair), or Class D (Poor). The classification is relative to the biological state of streams having similar observations based on the national distribution and therefore, it is not considered to be an absolute outcome. Table 1 provides the equations for computing the biological indicators (TDI, BMI, and FAI) developed by NAEMP [64].

Algae are important indicators of water quality change or water pollution and are widely used to evaluate the health of rivers [65]. TDI estimates diatom conditions based on species abundance and sensitivity and evaluates the nutritional conditions of stream ecosystems [66]. It is computed by calculating the composition using weighted mean sensitivity (WMS) measurements and proportion of benthic diatom taxa (Table 1).

Benthic macroinvertebrates are important biotic components of the river ecosystem and a food source for fish. Since they are extremely sensitive to habitat disturbances, they serve as an important biological indicator [67]. They exhibit distinct differences in the cluster structure, such as the number of species and distribution of populations [68]. BMI represents the condition of macroinvertebrate communities in stream ecosystems and describes the state of the communities based on habitat and environmental changes. The number assigned to each species, unit saprobic values, and frequencies are used as weight indicators for the species [69].

Fish as a biological indicator reflect the long-term environmental impacts within an aquatic ecosystem [70]. NAEMP developed the FAI using eight evaluation metric models (M1–M8) by analyzing the ecological characteristics of Korean fish communities. These metrics can be divided into four categories: composition of species, nutritional composition, fish abundance, and individual health [62,71].

### 2.3. Measurements and Selection of Scale

To measure the area of green and urban areas in the Nam-Han River Basin, we used land use land cover (LULC) data provided by the Korean MOE. The urbanized class was represented by “Urban,” while the forest and grass classes were represented by “Green.” The respective areas of each class were extracted using ArcMap software version 10.1. The areas were transformed into grids using the buffer features; additionally, the ratio of areas in each buffer was calculated as PLAND using the spatial pattern metric computing program Fragstats version 4.2 (The University of Massachusetts, Amherst, US) [72]. PLAND quantifies the area (in percentage) of each patch to the entire buffer and ranges between 0 and 100. It is a measure of landscaping composition that indicates the range of landscape with certain patches [73,74]. In addition, the DEM (digital elevation model) acquired from the Han River Flood Control Office and the National Geographic Information Institute was used to extract data on topographical variables in the Nam-Han River Basin for calculating the average elevation and slope for each buffer area.

In this study, two buffer scales were selected based on the sampling points to investigate the impact of green and urban areas and topographical variables on biological indicators. Multiple different scales have been used in previous studies [38,75]. Generally, the effectiveness of the riparian buffer improves with an increase in the buffer width, but various landscaping indicators have been reported to have different effects at different scales [76]. Furthermore, different maximum buffer sizes have been proposed based on previous research [77]. Accordingly, we selected (i) circle buffers with a 5 km radius from the sampling points and (ii) linear buffers of 500 m from the riparian areas to compare the large-scale effects of variables near the river (Figure 2). 

### 2.4. Spatial Autocorrelation and Data Analysis

Spatial autocorrelation is commonly measured by methods that use a single exponent, such as Moran’s *I* (MI, Equation (1); where *n* is the number of data, *X* is the pixel value, $\bar{X}$ is the mean of pixel values within the neighborhood range, and *W_ij_* is the spatial weight matrix) and Geary′s C, and methods that use variograms [78]. The single exponent method is useful for measuring the overall spatial autocorrelation of the study site, whereas variograms are useful for understanding the spatial distribution and structure of the autocorrelation [54,79]. In this study, we used MI to verify the spatial autocorrelation of the biological indicators of the Nam-Han River watershed. The MI values for the entire region were calculated using GeoDa version. 1.18 designed by Luc Anselin [80]. The theoretical range of correlation coefficients for MI is between −1 and 1, where coefficient values closer to 1 or −1 indicate a higher spatial autocorrelation, while a value closer to 0 indicates low spatial autocorrelation [81]. Spatial autocorrelation in the observed data can be corrected by separate sampling at a certain effective distance or by performing statistical corrections [82,83]. However, sampling at effective distances is practically difficult. Therefore, in this study, we conducted an analysis that allowed us to consider spatial autocorrelation as a statistical correction.
(1)$$I=\frac{n\sum_{i=1}^{n}\sum_{j=1}^{n}W_{ij}\left(X_{j}-\bar{X}\right)}{S_{0}\sum_{i=1}^{i=n}{\left(X_{i}-\bar{X}\right)}^{2}}$$

Principal component analysis (PCA) was performed to identify the correlation between the variables. PCA is frequently used as a preliminary test prior to performing other statistical analyses, and not as a final analysis method. Furthermore, the relationship between the variables can be graphically visualized, facilitating the identification of correlation. Moreover, normalization can rescale the data to identify the weights of the principal components (PCs), thus allowing the reduction of the variable dimensions based on the accumulation of the explanatory ability for critical variables [84]. In this study, we conducted PCA to estimate the relationship between the biological indicators, topographical variables, and PLAND of green/urban areas in the Nam-Han River Basin using the “prcomp” and “PCA” packages in R statistical language. PCA biplots provide a graphical visualization, describing the interrelationships between multiple datasets using vectors. The angle and direction of vectors indicate correlations between the original dataset and PCs; if the vector is highly parallel to the PCs, the impact (i.e., correlation) on the particular PC is high. PCA analysis is often used because it provides excellent visualization of the variables from multiple perspectives in the form of two-dimensional graphs [85].

Thus, since MI values extracted from biological indicators have a strong spatial autocorrelation, we conducted regression tree analysis, which is a statistical correction method for dependent samples. It is useful to acquire a sufficient number of samples devoid of spatial autocorrelation to facilitate easy interpretation [86,87]. Compared to usual linear models, the regression tree provides better prediction because it is appropriate for analysis of complex ecological data that can include nonlinear relationships between various variables, missing values, and more complicated interactions [83,88]. Regression trees with having hierarchical structures explain variations of responding variables by splitting predictor variables at certain thresholds minimizing the variance of responding variables. Regression tree analysis has been widely used in ecological studies [83,89,90,91,92,93]. The tree of the model grows repeating binary splitting of the data, and each split that is defined based on a single explanatory variable forms two nodes [94,95]. Selecting the size of the tree is performed through cross-validation, and the selected tree has the smallest predicted mean square error [96]. Therefore, in addition to PCA, we used regression tree analysis to maintain the maximum number of observations and solve the spatial autocorrelation problem inherent in the dataset. Additionally, the regression tree can provide a foundation for step-by-step examination of the effects of variables and assist decision making, unlike previous studies that used regression or simply correlation analysis. We used the “rpart” package along with ANOVA (analysis of variance) in R statistical language and generated one tree for each of the three biological indicators to identify its specific relationship with the variables of the Nam-Han River Basin. According to Everaert et al. [95], different parameterizations (e.g., the number of cross-validations, pruning, and the minimum number of observations per leaf) in regression tree analysis can result in considerably different outcomes. However, parameters were not specified in regression tree analysis. We selected the models that minimize ‘rel error’ (i.e., error on the observations used to estimate the model) and ‘xerror” (i.e., error on the observations from cross validation data) without overfit issue (i.e., simplest model) [97] (see Everaert et al. [95], for a more detailed description on how regression trees can vary due to different parameter settings in regression tree analysis).

## 3. Results

### 3.1. Descriptive Statistics

Table 2 provides a summary of descriptive statistics for the biological indicators, including the mean values of topographical variables, such as slope and elevation, and green/urban areas in the riparian areas within the 5 km circle buffer and 500 m linear buffer. The minimum values of TDI, BMI, and FAI were 11.20, 22.60, and 21.90, whereas their maximum values were 95.20, 94.70, and 100.0, respectively, suggesting that the biological conditions of the Nam-Han River show variability based on different sites within the basin. The standard deviation of FAI was 18.57, which was slightly lower than that of TDI and BMI, indicating marginal changes in FAI at the sampling sites. Furthermore, the standard deviation of TDI (19.78) was the highest among the three indicators, thus, suggesting relatively substantial changes in the diatom status.

The health score of the three biological indicators was defined by NAEMP as shown in Table 3 [64]. The mean values of TDI, BMI, and FAI indicated that the health score of the biological indicators in the Nam-Han River basin, were 56.08, 65.53, and 55.45, respectively (Table 2). Thus, according to the classification criteria defined by NAEMP, the condition of diatoms, macroinvertebrates, and fish in the Nam-Han River basin was “Fair,” “Fair,” and “Poor.” FAI showed the lowest mean value, whereas the mean value of BMI was the highest.

Moreover, the mean value of urban areas (14.56) in the 5 km circle buffer was significantly lower than that of green areas (50.10), suggesting a higher proportion of forests and grasslands than urban areas in the Nam-Han River Basin. The mean values of urban and green areas for 500 m linear buffers were 13.85 and 41.07, respectively, although the mean value of urban areas did not differ significantly depending on buffer scale. The proportion of green areas was higher than that of riparian areas since the mean value of the 5 km buffer was greater than that of the 500 m buffer. The mean values of topographical variables for 5 km and 500 m buffer zones were 212.00 and 158.92, respectively, for elevation and 10.07 and 7.85, respectively, for slope, indicating higher elevation and steeper slope of the terrain in the wide range buffers than in the riparian area.

### 3.2. Principal Component Analysis

PCA was conducted to examine and describe the degree of variation of the variables by the newly developed PCs (Table 4). Additionally, PCA presents the variables that have the greatest influence on each PC. Positive and negative signs denote direction, and the influence of variables is dependent upon the absolute value of the coefficient (Table 5). For example, PC1 represents the first PC, which can be expressed as a linear combined equation of the following variables:**PC1** = (TDI × 0.70) + (BMI × 0.83) + … + (Elev_50 m × 0.85) + (Slope_500 m × 0.93).(2)

PC1, which represented topographical variables, such as elevation and slope, accounted for 50% of the total variation, whereas PC2, which represented the proportion of urban and green areas at a 5 km buffer, accounted for 15.87% of the total variation. Thus, the cumulative explanatory power of the dataset by PC1 was 65.87% (Table 4). If the weight of a PC is highly directed to the positive direction, it indicates good biological condition. Conversely, if weight is highly directed to the negative direction, it represents poor condition. In PC1, all topographical variables a high weight. However, in PC2, the proportion of urban and green areas within the 5 km buffer exhibited the highest weight in the opposite direction. Furthermore, the proportion of urban and green areas within the 500 m buffer represented the highest weight in PC3 (9% of the total variance) and PC4 (7% of the total variance), and 82% of the dataset were explained by PC1–PC4.

Figure 3 is a biplot of the PCA that presents an effective visualization of the PCs. The x-axis of the biplot denotes Dim1 (primary PC), and the y-axis denotes Dim2 (secondary PC). Variables in the same direction are seen to be corresponding variables. The graph briefly represents the weights of the variables that affect PC1 or PC2, and their corresponding spatial locations indicate similarities between them. Thus, the figures show that the biological indicators (TDI, BMI, and FAI) and topographical variables (elevation and slope) represent similar trends, and urban and green areas represent perpendicular (high and low) trends. In particular, except for the topographical variables, the green area in the 500 m buffer was the closest related variable to biological indicators because the vectors were parallel to the PC axis as a closer distance indicates a stronger impact. Meanwhile, differences in the buffer scale were observed to be perpendicular to the vertical axis. Green and urban areas in the 5 km buffers had a stronger effect than 500 m buffer areas in PC2. Additionally, green areas exhibited a positive relationship with the three biological indicators, whereas urban areas were negatively associated in the 500 m buffer. However, the directionality of the 5 km and 500 m scales in urban areas differed as the urban areas did not have a significant impact on the biological indicators at a wider range. Therefore, the analysis focused on identifying the effect of the impact of the relationship on biological indicators at each scale and type of area.

### 3.3. Spatial Autocorrelation Verification of Biological Indicators

Analysis of the spatial autocorrelation of TDI, BMI, and FAI in the Nam-Han River Basin using GeoDa indicated high spatial autocorrelation of the biological indicators (Figure 4). In particular, the spatial autocorrelation coefficient (MI) for BMI was the highest (0.834), which was followed by that of FAI (0.785) and TDI (0.779). The topographical and land use variables and biological indicators showed MI values of 0.5 or higher, indicating a significant spatial dependency of the variables (*p* < 0.01) [98]. The high spatial autocorrelation of the biological indicators indicates that the spatial distribution of stream biological conditions depends on the hierarchical structure of environmental conditions [54]. Thus, the stream biological condition at any point is not entirely independent and instead, depends on the biological health around specific locations. These results suggest that conducting sampling and statistical analyses without considering spatial autocorrelation of stream biological indicators can cause potential errors.

Furthermore, the MI values estimated in this study do not only represent the spatial autocorrelation or dependency of the measurements across the variables but also clustering or dispersibility of the variables. The values represent a random distribution close to zero, strong clustering close to +1, and strong dispersion of uniform close to −1; therefore, the high MI values of the three biological indicators of the Nam-Han River Basin indicate that their conditions are clustered [81].

### 3.4. Regression Tree Analysis

The slope of the 500 m buffer was the first variable to determine the differentiation for the biological indicators of the Nam-Han River Basin. This was consistent with the PCA results, which indicated that the slope of the 500 m buffer had the highest weight among the first PCs (0.93). However, the biological indicator results differed after Node1.

The TDI regression tree indicates that the slope within the 500 m linear buffer in the riparian areas is the most significant variable to describe TDI (Figure 5). The observations were divided into two groups according to the slope, wherein groups with a slope less than 7.6% denoted a poor TDI condition, while groups with slopes more than 7.6% indicated a good TDI condition. The elevation of the 5 km circle buffer was the second most important variable to describe the TDI score in the group with a slope less than 7.6% (left side of the tree). A high elevation increased the mean TDI value by 21% and was associated with a good status of diatoms. In areas with slopes of 7.6% or higher (right side of tree), the proportion of green areas in the 500 m linear buffer was the second most important variable to explain the TDI status. In areas with relatively high slopes, a high proportion of green areas near the riparian system indicated better biological conditions of diatoms. After the mean elevation of the 5 km buffer, the green area in the 500 m buffer and 5 km buffer represented the third important variable on the left and right sides of the tree, respectively. The R^2^ value of the regression tree for TDI was 0.67, and the relative importance of the variables was 20% (500 m slope), 18% (5 km slope), 16% (5 km elevation), 15% (500 m elevation), 13% (500 m green area), 12% (5 km green area), 3% (500 m urban area), and 2% (5 km urban area). Other than the topographical variables of the stream, an increased proportion of the forest and grass area near the riparian area, represented by the 500 m buffer green rate, improved the condition of diatoms.

The first differentiation in the BMI regression tree was also the slope within 500 m of the riparian area (Figure 6). In the lower slope group (left of the tree), the second most important variable was the green area in the 5 km circle buffer, which was again classified as the urban area of the 500 m buffer at the divided node. However, the second most important variable in the high slope group (right side of the tree) was elevation. Low values of elevation in the 5 km buffer were affected by the BMI condition in the green area of the 500 m buffer. Furthermore, the wide range of green areas with lower slopes (i.e., relatively less fluid flow) had a significant impact on the condition of macroinvertebrates, suggesting that the green area near the stream was comparatively more important among the areas with high slopes (i.e., areas with more fluid flow). The R^2^ value of the BMI regression tree was 0.78, and the relative importance of the variables was 20% (500 m slope), 15% (5 km elevation), 14% (500 m elevation), 14% (5 km green area), 13% (500 m green area), 3% (500 m urban area), and 2% (5 km urban area). Similar to the TDI results, the proportion of the green areas at wide ranges, represented by the proportion of green areas within the 5 km buffer, influenced the improvement of macroinvertebrate conditions along with the stream topographical variables.

In the FAI regression tree, fish condition scores were divided first by the 500 m slope (Figure 7). Observations with slope less than 9.3% formed a second node at the urbanization rate of the 5 km circle buffer. For FAI values classified in high urbanization areas (left of the tree), the third most important variable was the slope of the 500 m buffer, which was then split into the elevation and urbanization rates of the 500 m buffer at the next node. In groups with a high slope of 9.3% or more (right side of the tree), the second most important variable that determined fish condition was elevation of the 500 m buffer, which indicated the mean elevation of the riparian area from the sampling point. Unlike the TDI and BMI regression trees, the FAI regression tree indicated that along with the topographical variables, the fish condition was more likely affected by the urban areas rather than green areas. In the low slope areas, the urban area within the 5 km buffer determined the biological status of fish. Additionally, in the high slope areas, the impact on the surrounding land use and fish condition was determined only by the elevation near the streams. This suggests that the biological conditions of fish in the Nam-Han River Basin were more likely affected by pollutants from urban areas around the stream than by the surrounding forests. The R^2^ of the regression tree for FAI was 0.72, and the relative importance of the variables was 21% (500 m slope), 17% (5 km elevation), 17% (5 km slope), 13% (5 km urban rate), 10% (5 km green rate), 4% (500 m urban rate), and 1% (500 m green rate). Along with the stream topographical variables, the proportion of urban areas in a wide area, represented by the 5 km buffer urbanization rate, influenced the improvement of the fish condition.

## 4. Discussion

### 4.1. Spatial Autocorrelation of Stream Biological Indicators

Previous studies investigated the biological indicators of stream condition without considering the spatial dependency inherent from the systemic structure of stream ecosystems [99,100,101]. Such inherent spatial dependency of variables violates the basic assumption of the conventional statistical analyses and may under/over-estimate the LULC in watershed and riparian areas based on the physiochemical properties and ecological condition of streams [48]. In this study, the three biological indicators (i.e., TDI, BMI, and FAI) in the Nam-Han River showed significant spatial autocorrelation.

Spatial dependence in stream biological communities can occur due to multiple reasons. By the definition, streams are a fluid ecosystem in which water continuously flows from headwater in mountainous areas to the estuary in lowlands which is the heart of river continuum concept [102]. Thus, polluted or ecologically impaired headwater can directly affect conditions downstream, despite the concentration of pollutants and the degree of impairments can be mediated or more deteriorated by other tributaries. Therefore, the physiochemical and biological conditions of streams are dependent on the condition of the headwater and the feeding streams resulting in spatial autocorrelations between upstream and downstream at various spatial scales [103,104,105]. Beside the nature of stream itself, the most common probable cause of spatial dependency in stream properties is topographic characteristics (e.g., elevation, and slope) in watersheds. Topographic features are not randomly distributed and have structural patterns in space. Thus, the topographic characteristics of a certain location are dependent on surroundings [106,107]. In our study areas, we were able to observe significant spatial autocorrelations of elevation and slope. In literature, topographic characteristics have been shown to determine the hydrologic and hydraulics systems, the amount of pollutant loading, run-off process, and pollutant delivering mechanisms [108,109]. Topographic characteristics even affect the availability, types, and intensity of land uses in watersheds and riparian areas [110]. Thus, it is evident that stream systems in the watershed are strongly tied with topographic characteristics in various ways. We also observed strong correlations of elevation and slope with the amounts of forests and biological indicators including TDI, BMI, and FAI through PCA. Besides, slope and elevation appeared as the first and the second splitter in our estimated regression tree models for TDI (Figure 5), BMI (Figure 6), and FAI (Figure 7), which emphasize the importance in explaining the variances of TDI, BMI, and FAI across the study areas. Since the topographic characteristics are structured systems over space (i.e., spatially autocorrelated), and stream systems are strongly tied with topographic systems, the spatial dependency of physiochemical attributes and biological communities of streams are inevitably inherent from spatial autocorrelation of topographic properties. Numerous studies argued the problems using conventional statistical methods (e.g., Spearman’s rank correlation and ordinary least square regression) assuming independency among observations [111,112]. Our findings suggest that spatial autocorrelation should be considered while assessing the relationship between environmental variables and stream quality.

### 4.2. Land Use and Topographical Effects on Biological Indicators of Stream

Diatoms, macroinvertebrates, and fish, as indicators of stream condition, exhibit different characteristics and complex mechanisms that interact with environmental and topographical variables [113,114]. According to the PCA and regression tree analyses results, topographic variables, including elevation and slope, had the strongest relationship with biological indicators, and the relationships between urban areas, green areas, and biological indicators at the 5 km scale were stronger at those at the 500 m scale. However, the variables that determined the second differentiation differed on the regression trees of the three biological indicators. The regression trees showed that slope at the 500 m scale had the greatest effect on the biological indicators. In the case of the TDI regression tree, the percentage of green areas at the 500 m scale was the most important factor after topographical characteristics. However, the BMI regression tree indicated that the green area at the 5 km scale had strong effects on the biological indicators at low slopes, while the green area at the 500 m scale had strong effects on the biological indicators at steep slopes. Moreover, the urban area at the 5 km scale had greater effects on FAI than at the 500 m scale.

The results of this study can be discussed in two aspects. First, the reason why the initial split was estimated as a topographical variable is that because of most of the tributaries in the Nam-Han River Basin are in forests. Furthermore, topographical factors, including elevation and slope are important variables for determining the biological indexes [17]. The forests are at high elevation and have a steep terrain; therefore, elevations and slopes are critical for determining the spatial characteristics of the stream ecosystem because they represent structural determinants of the amount and speed of water flow [115]. Subsequently, the amount and speed of stream water can significantly change the biological condition in the stream. In addition, elevation and slopes may restrict land use around streams. Urban and agricultural areas that are majorly built downstream at low elevation and slope, greatly influence the stream biological indicators [116]. Therefore, the highest elevation and slope variables affect the biological condition of the tributaries in the Nam-Han River Basin since forests comprise 70% of the total basin area. Second, the biological characteristics of each biological indicator are different. Diatoms are largely affected by the surrounding regions [117]. They are sensitive to light and water temperature, and that is greatly influenced by its conductivity. Urbanized rivers generally have a higher conductivity than the rivers that flow through forested landscapes. This increases the intensity of light and increases the water temperature, making it a habitat disadvantage to the species [118]. Consistent with the results of this study, topographical variables and green areas in the riparian areas were critical for diatoms. Furthermore, macroinvertebrates and fish respond distinctly to changes in their physical habitats. This is more evident for fish, which directly represent stream conditions since they are the top predator in the food chain of the stream ecosystem [119,120]. According to studies by B. Schmalz et al. [121], macroinvertebrates have the strongest correlation with dominant substrates, and fish have a stronger correlation with flow variability and immediate land use. Additionally, another prior study of this ascribed greater predictive power to land use by comparison the covered a larger area, providing greater contrast among sub-catchments [122]. The results concluded that the importance of local habitat conditions is best revealed by comparing at the within sub-catchment scale. Thus, it is assumed that macroinvertebrates were most affected by proportion of green areas on a wide-area scale, which is the largest land use in this study. And except for the topographical variables, the proportion of urbanized area is acting as the biggest factor in the species status change of fish because the changes in land use are relatively immediate. Since the detailed life cycle, food chain and movement mechanisms of the three biological indicators are different, they have different environmental variables that are most affected by each of them, even in the same site [123,124].

## 5. Conclusions

This study examined the relationship between land cover (represented as green and urban areas) and biological indicators according to two buffer scales. Subsequently, different variables were the first differentiation criterion for the regression tree of each biological indicator, thus indicating that the effect of green and urban areas differed and depended on the condition of each biota in the stream. Additionally, the effect of the riparian proximity difference was compared by comparing the proportion of areas in the 5 km circle and 500 m linear buffers. Our findings indicate that these effects can vary significantly depending on the topographical variables of the watershed. Furthermore, as the topography of the study area was influenced by the characteristics of the forest terrain in the Nam-Han River Basin, considering the river topography during land use planning is essential in the basin.

This study provides several implications for planning and managing LULC in watershed and riparian areas. First, the study emphasizes that spatial autocorrelation originating from the spatial characteristics of river ecosystems along with their relationships with land use and topographical characteristics should be considered while analyzing the biological condition of the stream. Subsequently, this could provide additional information, such as the criteria of node splitting, and correlation of various effects on river ecosystems. Second, our results provide a basis for considering the impact of locations of green and urban areas on each biological index through a comparison on two different scales. Positive effects of green areas are commonly known, but sufficient literature on the effect of the spatial extent of different types of areas on each biological indicator at the same study site does not exist. Therefore, this study verified and compared the difference in the impact of the location of green and urban areas on each species. However, the present study was limited to green areas (including forests and grasslands) and urban areas (including cities, roads, and buildings) in the Nam-Han River Basin. Thus, future research should include other environmental characteristics of the basin. Third, the regression trees that were constructed for the biological indicators can assist in establishing and prioritizing stream restoration strategies by determining the significance of topography and land cover variables for the condition of each biological indicator in the basin. Different land cover types and topographical characteristics affect different biological indicators, which in turn indicate the biological condition of the basin by determining the unhealthy indicators. Thus, the results of this study can serve as a baseline for supplementing efficient management of the basin. However, arguably the most significant limitation of regression tree analysis is that it is a nonparametric approach which makes it hard to generalize the analysis results based on the observed data. The estimated regression tree model is a black-box model simply describing the relationship between predictors and responses [95] and is built based on the observed datasets without inference about the underlying probability distribution [125]. Another limit of the regression tree analysis is the complexity of the estimated regression tree when it has many nodes, and, in such cases, interpretation of the estimated tree can be challenging [95,126,127]. Therefore, additional studies should be conducted to investigate these limitations further and to develop reliable tools for moderating these issues. 

## Figures and Tables

**Figure 1 ijerph-18-05150-f001:**
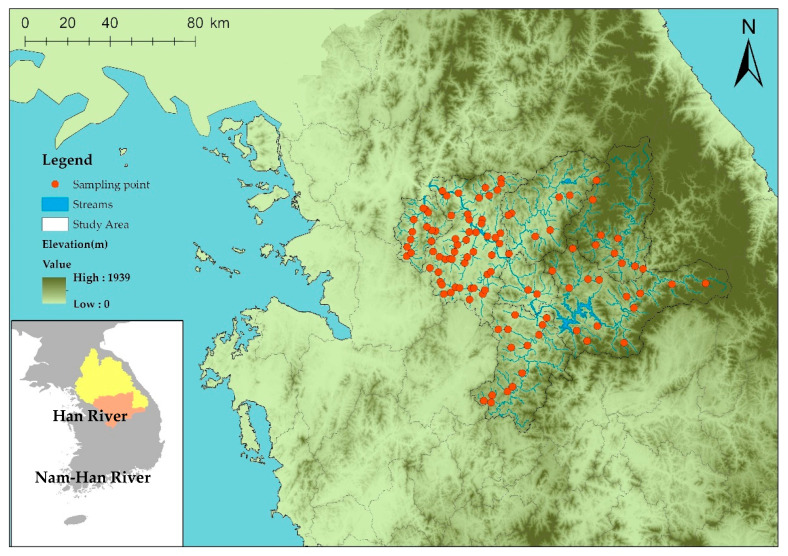
Sampling sites in the Nam-Han River Basin in Korea selected by the National Aquatic Ecological Monitoring Program.

**Figure 2 ijerph-18-05150-f002:**
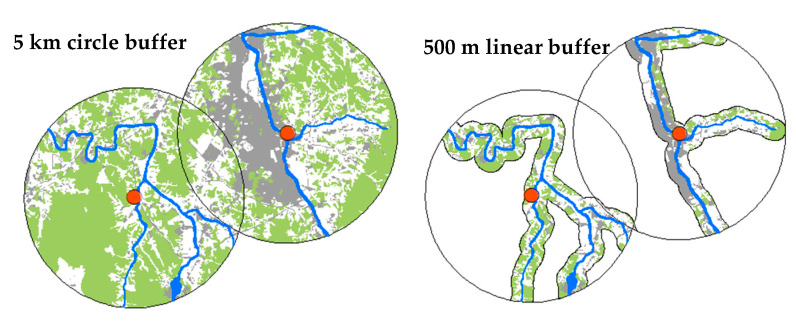
Circle buffer scale set to 5 km radius in the watershed and linear buffer set to 500 m in the riparian area near the sampling sites (green color; forest and grass areas, gray color; urbanization areas, blue color; streams, orange circle; sampling points).

**Figure 3 ijerph-18-05150-f003:**
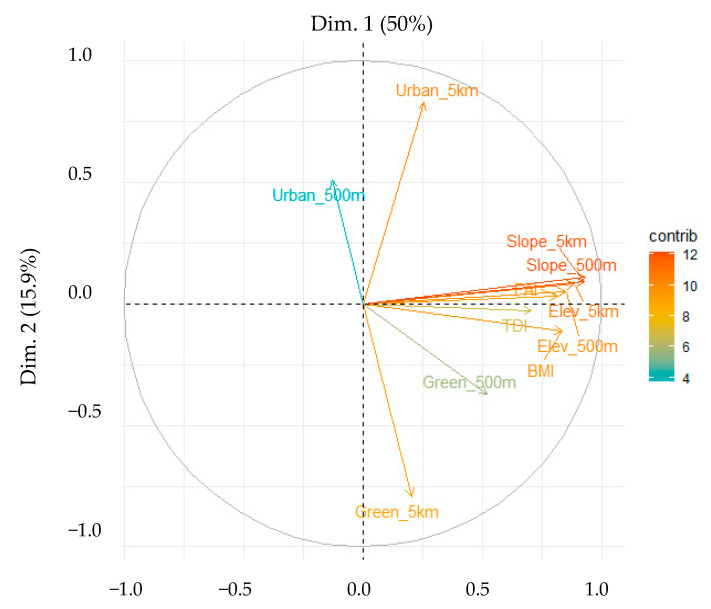
Principal component analysis biplot for the first two principal component scores.

**Figure 4 ijerph-18-05150-f004:**
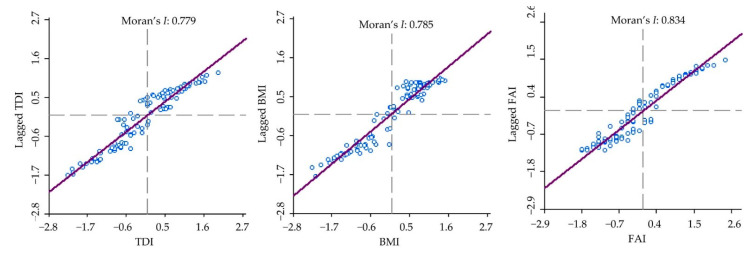
Moran’s *I* values of the three biological indicators.

**Figure 5 ijerph-18-05150-f005:**
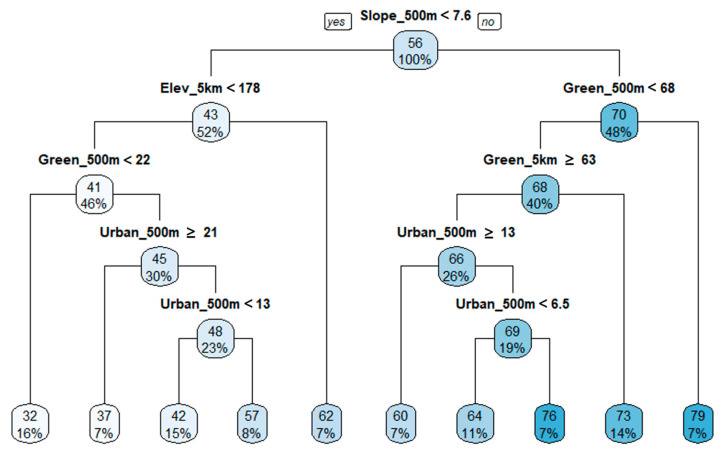
Regression tree for TDI with stream topographical variables and proportions of green and urban areas in buffer areas (R^2^ = 0.67). The numbers located on top in color boxes are mean values of TDI.

**Figure 6 ijerph-18-05150-f006:**
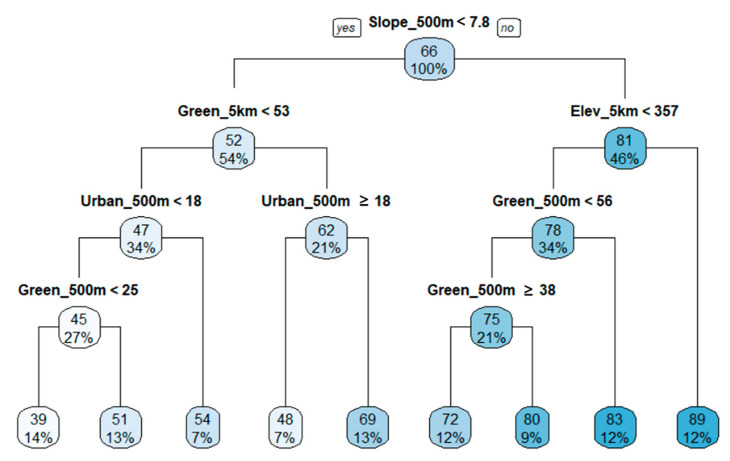
Regression tree for BMI with stream topographical variables and proportions of green and urban areas in buffer areas (R^2^ = 0.78). The numbers located on top in color boxes are mean values of BMI.

**Figure 7 ijerph-18-05150-f007:**
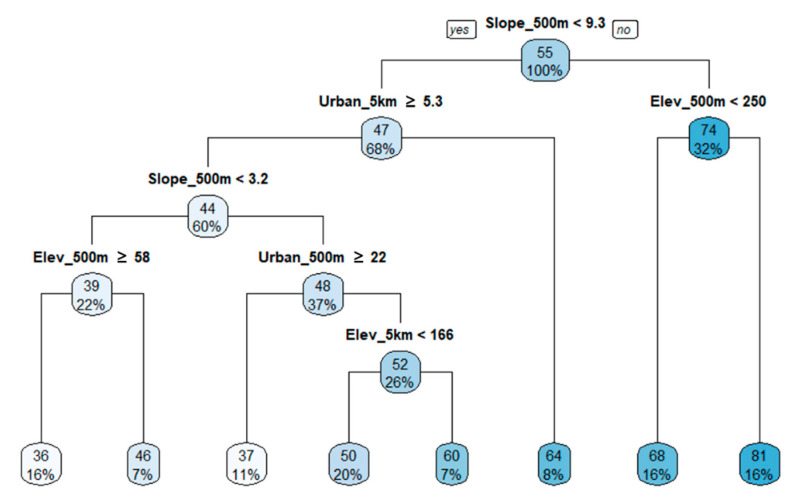
Regression tree for FAI with stream topographical variables and proportions of green and urban areas in buffer areas (R^2^ = 0.72). The numbers located on top in color boxes are mean values of FAI.

**Table 1 ijerph-18-05150-t001:** Equations for computing the biological indicators developed by National Aquatic Ecological Monitoring Program (NAEMP).

Biological Indicators	Equations
TDI (Trophic Diatom Index)	TDI = 100 − {(WMS × 25) − 25} WMS: weighted mean sensitivity $WMS=\sum^{\;}A_{j}\cdot S_{j}\cdot \frac{V_{j}}{\sum^{\;}A_{j}\cdot V_{j}}$ Where, j = species *A_j_* = abundance (proportion) of species j in the sample (%) *S_j_* = pollution sensitivity (1 ≤ *S* ≤ 5) of species j *V_j_* = indicator value (1 ≤ *V* ≤ 3)
BMI (Benthic Macroinvertebrates Index)	$\mathrm{BMI}=\left\{4-\overset{n}{\underset{j=1}{\sum }}K_{j}H_{j}G_{j}/\overset{n}{\underset{j=1}{\sum }}H_{j}G_{j}\right\}\times 25$ where, j = number assigned to species n = number of species *K_j_* = unit saprobic value of species j *H_j_* = frequency of species j *G_j_* = indicators weight value of species j
FAI (Fish Assessment Index)	FAI = sum of 8 metrics Metric 1 (M1): number of Korean native species Metric 2 (M2): number of rifle benthic species Metric 3 (M3): number of sensitive species Metric 4 (M4): percentage of tolerant species Metric 5 (M5): percentage of omnivores Metric 6 (M6): percentage of insectivores Metric 7 (M7): the amount of collection native species Metric 8 (M8): percentage of fish abnormalities

**Table 2 ijerph-18-05150-t002:** Descriptive statistics of biological indicators, topographical features, and green/urban areas in the Nam-Han River Basin.

Classification	Variables	Min.	Max.	Mean	S.D.
Biological indicators	TDI	11.20	95.20	56.08	19.78
	BMI	22.60	94.70	65.52	19.19
	FAI	21.90	100.0	55.44	18.56
Topographical features	5 km Mean elevation (m)	48.85	764.73	212.00	139.73
	500 m Mean elevation (m)	38.90	589.63	158.91	109.00
	5 km Mean slope (%)	1.99	23.29	10.06	4.91
	500 m Mean slope (%)	1.64	21.54	7.85	4.87
Green spaces	5 km Forest and Grass area (%)	4.58	90.91	50.10	23.87
	500 m Forest and Grass area (%)	4.82	78.21	41.06	19.73
Urban areas	5 km Urban area (%)	0.98	91.74	14.56	19.48
	500 m Urban area (%)	2.33	66.06	13.85	12.52

**Table 3 ijerph-18-05150-t003:** Classification of biological indicators according to NAEMP.

Biological Conditions	Class	TDI	BMI	FAI
Good	A	60 ≤ TDI ≤ 100	80 ≤ BMI ≤ 100	87.5 ≤ FAI ≤ 100
Fair	B	45 ≤ TDI < 60	60 ≤ BMI < 80	56.2 ≤ FAI < 87.5
Poor	C	30 ≤ TDI < 45	45 ≤ BMI < 60	25 ≤ FAI < 56.2
Very Poor	D	0 ≤ TDI < 30	0 ≤ BMI < 45	0 ≤ FAI < 25

**Table 4 ijerph-18-05150-t004:** Proportion of variance and cumulative proportion for each principal component.

Variable	PC1	PC2	PC3	PC4	PC5	PC6	PC7	PC8	PC9	PC10	PC11
Proportion of Variance	0.50	0.15	0.09	0.07	0.06	0.03	0.02	0.02	0.02	0.00	0.00
Cumulative Proportion	0.50	0.65	0.75	0.82	0.88	0.92	0.95	0.97	0.99	0.99	1.00

**Table 5 ijerph-18-05150-t005:** Rotated factor matrix extracted from principal component analysis. The underline values are the largest weighs on that PCs.

Variable	Principal Component				
	PC1	PC2	PC3	PC4	PC5
TDI	0.70	−0.02	0.11	0.48	−0.28
BMI	0.83	−0.11	0.14	0.16	−0.21
FAI	0.81	0.03	0.01	−0.07	−0.35
Urban_5 km	0.25	0.82	−0.21	0.17	0.18
Green_5 km	0.20	−0.79	0.39	−0.12	0.19
Elev_5 km	0.88	0.07	−0.15	−0.32	0.16
Slope_5 km	0.92	0.09	0.14	−0.02	0.10
Urban_500 m	−0.12	0.50	0.78	0.05	0.24
Green_500 m	0.52	-0.37	−0.28	0.49	0.47
Elev_500 m	0.85	0.05	−0.15	−0.39	0.13
Slope_500 m	0.93	0.10	0.14	−0.03	0.01

## Data Availability

Not applicable.

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
