# Peer review of "Regression Tree Analysis for Stream Biological Indicators Considering Spatial Autocorrelation"

_ijerph, 2021, doi:10.3390/ijerph18105150_

Round 1

Reviewer 1 Report

The paper explains the relationships between land cover and biological indicators, focused on planning and management. In my opinion, the study has been well-planned and developed, the text is correctly written, and the information is orderly and clearly exposed. I am not an expert on statistical and regression tree analysis and spatial autocorrelation, but I believe this approach is consistent. I think that this study is worthy of being published; only a few minor changes should be considered:

Line 85: delete ‘using Moran’s I (MI)’ and introduce this abbreviature in the Material & Methods section.

Table 1: Regarding FAI (Fish Assessment Index), is the sum correct? The index is the sum of numbers and percentages, without any kind of calibration?

Lines 182-186: This is not Material & Methods; please replace this on Discussion.

Reviewer 2 Report

I have carefully read the manuscript and it seems to me that it presents the quality that the journal needs. I think that the introduction provides sufficient background and presents the relevant references; the methods are adequately described. Results are clearly presented and discussed, and the conclusions are supported by the results. For all this, I recommend its publication in the present form.
